# Multi-User Virtual Reality for Remote Collaboration in Construction Projects: A Case Study with High-Rise Elevator Machine Room Planning

**Phong Truong** [1,2,*] **, Katja Hölttä-Otto** [1,3] **, Paulina Becerril** [2] **, Reetta Turtiainen** [2] **and Sanni Siltanen** [2]

1   School of Engineering, Aalto University, 02150 Espoo, Finland; katja.holtta-otto@aalto.fi
    or katja.holttaotto@unimelb.edu.au
2   KONE Technology and Innovation, Myllykatu 3, 04830 Hyvinkää, Finland;
    paulina.becerrilpalma@tuni.fi (P.B.); reetta.turtiainen@hel.fi (R.T.); sanni.siltanen@kone.com (S.S.)
3   Department of Mechanical Engineering, University of Melbourne, Parkville, VIC 3010, Australia
*   Correspondence: phong.truong@aalto.fi

**Abstract:** Virtual Reality (VR) is considered among the major technologies to address the inefficient collaboration issue caused by the predominant use of 2D drawings in the construction industry. However, there is still a knowledge gap between researchers and practitioners about the actual benefits of VR in the business context. This paper presents the benefits of VR usage in four real-life high-rise elevator projects from the user and business perspectives. Four VR environments of actual machine rooms for planning were created and tested in a multi-user setting. Overall, users find VR more intuitive than conventional tools to enhance planning accuracy and collaboration. The results also show that VR brings significant economic savings and gains for business in the industry. Future study should investigate the real cost-benefit ratio of VR and streamline its technical implementation within construction projects. The research contributes to the current body of knowledge by providing real-life economic benefits and directions to address the research gap in both academia and industry to promote the wide adoption of VR.

**Keywords:** multi-user virtual reality; remote collaboration; construction planning; user need; economic benefits; VR; collaborative VR; virtual reality

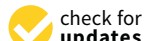



## 1. Introduction

The unique nature of the architecture, engineering, and construction (AEC) industry involves extensive communication between stakeholders regarding layout design, construction site logistics, facility requirements, etc. However, such communication in construction projects is often inefficient, which hinders the collaboration between them [1–3]. One of the main challenges is the dominant use of 2D drawings as a means of design communication throughout the project lifecycle. This media is considered as the major constraint and accounts for design issue overlook, delay, and ultimately cost overruns [2].

In this paper, we focus on one particular AEC scenario, elevator machine room planning for high-rise buildings. Such projects are complex; hence planning is a demanding task. As the machine room is unprofitable space, it is usually provided with minimum area but packed with equipment from both the elevator company and other contractors. This complicates the planning process. The layout design not only needs to fit all components inside but also needs to have enough clearance space for future maintenance activities. Common to the industry, 2D drawings are used as the main design communication method. However, the drawings often lack critical items such as trunking, cable routing and equipment from other stakeholders. This insufficient information has led to installation errors causing safety hazard, rework, delay, and equipment unreliability [2,4].

Moreover, remote communication is highly characteristic of construction projects [3,5]. Outsourcing complicated design to international vendors and having project teams with

diverse geo-location are common practices in the AEC industry [6,7]. Together with the impact of COVID-19, the shift toward remote collaboration is prominent now more than ever [8]. Besides, the engineering complexity of construction projects is constantly increasing, and at the same time when client's expectations for quality are rising. Hence, there is a need for a better means of communication in the AEC industry [2,5].

Virtual Reality (VR) is among major technologies to digitization in Industry 4.0 [2,9]. It is important to note that the term VR has multiple meanings [1,10–13]. This paper refers to VR as "the uses of computer science and behavioral interfaces to simulate the behavior of 3D objects in a virtual world, enabling real-time interactions with each other in pseudo-natural immersion via sensorimotor channels" [1,14]. Extending from this definition, multi-user VR is the VR technology that enables more than one user to simultaneously experience and interact with each other in a shared virtual environment [15–17].

The attention to VR has grown due to the rising of building information modelling (BIM) practice in the industry [18,19]. BIM-based VR has the great prospect in design visualization to better understand the design complexity and enhance its communication [1,2]. In addition, the cloud-based multi-user VR system that support several users at the same time has also gained more attraction thanks to its ability to support remote collaboration between stakeholders [20]. Nonetheless, the AEC industry has still slowly adopted the technology compared to other industries such as medical [21]. The knowledge gap between researchers and practitioners about the actual benefits of VR in the business context accounts for this issue [21,22].

This study investigates the use of multi-user VR in the elevator machine room planning process via the case study of four real-life high-rise projects. The aim is to understand the real-life benefits of VR in the industrial context and inform the remaining research gap to support future study in the field. Our main research question is to identify the actual benefits of VR from the user and business perspective in the AEC industry.

The paper is structured as follows: we first present related works to provide some background of the research field in Section 2; Section 3 details the research methodology and practical implementation; Section 4 describes the results of our study; Section 5 discusses our findings as well as limitation and recommendation for future study; and Section 6 presents our conclusion for this research.

## 2. Related Work

VR and collaborative VR have been an emerging research topic in the AEC academia and industry [1]. This section briefly presents the current usage of VR, its perceived and economic benefits, as well as the limitations that prevents it from being adopted in the AEC industry from the existing body of research.

### 2.1. The Use of VR in AEC Industry

In the design phase, VR has been used to communicate construction design in the AEC industry [1,2,9,17]. Studies have acknowledged design review as the major application of VR in the AEC industry [2,17,22,23]. Most importantly, such usages often adhere to user involvement rather than co-design [2]. In other words, the focus is on involving stakeholders in pre-construction and post-occupancy evaluation via building walk-through [2,24–26]. In most cases, design review in VR is utilized in complex projects such as airports and healthcare facilities [1,17,21]. Nevertheless, Whyte [27] also reports such uses in small projects where the VR environment can be reused in a study on how lead user firms in the construction sector in USA and UK used VR. Moreover, VR serves as a tool to research and train human behavior in the built environment as well as feed into the design process [1,17]. Major use cases are simulating an emergency such as fire and earthquake [28–31].

In the construction phase, VR has been utilized as a method to visualize field construction planning [32]. The urgency of using VR results from the inefficient conventional construction management using complex graph-based data for spatial-temporal planning [17]. Many studies have developed VR systems for construction scheduling,

discrete-events simulation, and on-site real-time tracking [33–35]. Moreover, construction safety planning, training, and monitoring in VR has gained more popularity in the AEC research community [1,36,37]. Safety training systems with different levels of realism from just a prompt-message to stimulation of other senses such as adding sound from the construction site has been developed in many studies [32,38–41]. Besides, multi-user VR has been utilized to provide collaborative training with simulated scenarios and even real-time tracking to enable monitoring [42–45]. Despite the proven benefits from research, VR usage for safety training in the industry remains low [22,46]. Augmented reality (AR), in this type of application, tends to have a wider adoption than VR [17,47].

### 2.2. The Perceived Benefits of VR

The perceived benefits of VR have been reported in many literature reviews [1,2,46,47]. These studies show that the benefits of VR depend on different use cases when applied properly. The interactive, spatial, and real-time properties of VR simplify the communication through immersive and full-scale design representation, in which user's movements are intuitive and non-restricted [15,19,48,49]. Instead of speaking from abstraction, VR offers a tangible frame of references to reduce and eliminate understanding gaps between visual thinkers (designers) and non-visual thinkers (other stakeholders) [1,19]. Even though CAD/BIM tools offer 3D-modelling, they are often too complicated for non-engineering stakeholders with limited spatial understanding [1,50].

Moreover, such immersiveness from VR brings a strong physical presence as "being inside the building" that may not be experienced in other media [19,28–30,51,52]. This physical presence results in the obtainment of non-verbal feedback which is signficantly beneficial in the construction design review process [1,17,53]. It can also induce real-life physical and psychological response that is critical and creates better safety training compared to conventional methods via documents [15,32,38–40,52,54].

Furthermore, multi-user VR is found most effective to connect remote stakeholders virtually [20]. This is critical as the number of virtual teams [7] and the need for remote work in the COVID-19 pandemic [8] has become a common practice in the AEC industry. Research has shown that the embodiment properties of multi-user VR can provide a digital equivalence to face-to-face communication in construction projects [3,15]. Such property evokes subtle but essential non-verbal information through movement of avatars and deictic references like "here", "this" and "there" [5,55].

### 2.3. The Economic Benefit of VR

Even though the economic benefits of VR are not comprehensively documented [21,25], three main categories can be identified. Firstly, direct cost and time savings (up to 90%) come from the use of virtual mock-ups to replace physical ones [22,56]. Such savings are reported in 93% of projects surveyed in a study on how contractors in the AEC industry used VR [22]. This source of savings also mostly accounts for the main economic benefits of VR throughout construction life-cycle (from planning to de-commissioning) [57].

Secondly, indirect savings results from improving design communication and collaboration as well as avoiding travelling need within construction projects. By using collaborative virtual walk-in to effectively detect unexpected design error, a British railway company has reported saving up to millions of pounds and speeding up the track improvement process [17]. As an response to national lockdown, one Singaporean consultant firm has saved four working days in a single BIM coordination task that often took five days before using multi-user VR [8]. Another firm in the same study reported saving S$ 100,000 out of S$ 400,000 operational expenses (64,000 out of 256,000 euros, according to the exchange rate on 22 June 2020) in 2 months. Besides, a reduction of two to three days of training to 45 min as well as the scheduling challenges for everybody to be at the same time and place is reported in the case of Siemens providing remote maintenance services via virtual mock-ups as the customer is off-shore [58]. This example also amplifies the future economic benefit of the re-usability of VR mock-up [27].

Thirdly, VR also brings business values in sales and marketing phases to AEC firms. As VR is still considered as novel in the AEC industry, adopting VR can result in the glossy image of companies that place innovation as a business value proposition [17]. For instance, BN Builders (an American construction company) has acquired more contract awards and Hexacon (a Singaporean construction company) has seen a 25% increase in sales revenue (S$50,000—approximately 32,000 euros, according to the exchange rate on 22 June 2020) in sales revenue after using VR [8,59]. With the marketing strategy to give potential buyers a VR tour of new apartment blocks, Permission Homes, a British housing developer, has attracted sales even when the construction has not begun [17].

*2.4. Current Limitation for Adoption in AEC Industry*

From a technical perspective, the interoperability issue between VR software and BIM data is the major limitation that hinders the use of VR in the AEC industry [1,22,36]. The transferring process of BIM into VR and vice versa is non-robust, which results from a lack of VR standard to define the technical implementation [21,60]. Most of common commercial VR software only focuses on reproducing geometry and texture while data such as cost, or element identification is not supported [20,21,25]. Direct transfer of data generated in VR to most common BIM tools are also not available, which requires extra manual work to integrate changes in VR into the original BIM model [23]. Moreover, the use of BIM in practice is limited even though they are deemed popular in the AEC industry [22,24]. This inhibits VR application, as it requires accurate BIM models to create the virtual environment [19].

The resistance of decision-makers to adopt VR in AEC companies is also another major limiting factor [19,21]. The lack of cost-benefit analysis and the VR familiarity gap between researchers and industry practitioners account for this challenge [21]. Hence, there is an increasing need for cost-benefit analysis and clear business use cases of VR [21,24,25]. This is essential to assist business to acknowledge the economic benefits of VR and how they could optimally adopt the technology to their workflow [24]. Upper management should also be educated about VR to increase their awareness which is found to be important in forming the willingness to use the technology in the AEC industry [19].

## 3. Methodology

The overall research procedure, summarized in Figure 1, was conducted with a user-centric mindset [61]. As placing user-centricity in research helps to capture the right human needs and co-create with the users [62,63], such approach fits our purpose in identifying the actual benefits of VR from the end user and business perspective. To acquire practical insights from the industry, four real-life high-rise elevators projects of KONE in the US, Indonesia, Dubai, and Malaysia were involved as the studied pilots.

All research activities were conducted remotely. In the first stage, key stakeholders were individually interviewed to identify the current challenges in machine room planning and their initial perception of VR. The insights generated were then fed into designing VR environments and user tests in the second stage. The creation of the VR environments was an iterative process. Researchers conducted bug fixings and improved some features on different versions until achieving the final design. In the last stage, participants performed the user test and joined a group interview. Due to the national COVID-19 lockdown in Malaysia at the time of the research, user tests involving the Malaysian pilot were cancelled. It is important to note that participants of the Malaysian pilot were still interviewed in the first stage. All collected data, which was qualitative from the individual interviews as well as the group interviews and our observation in the user tests, was analyzed using the Affinity Diagram method [64].

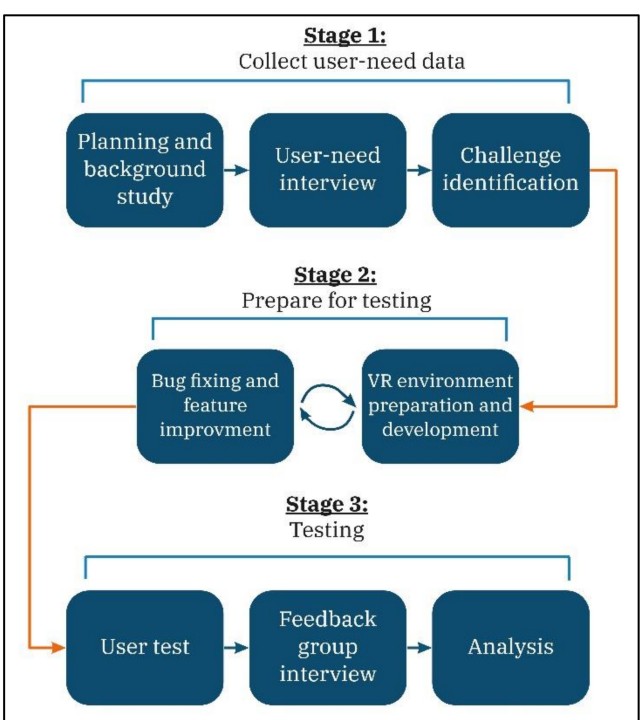

**Figure 1.** Overall research procedure.

There were 18 participants in total with 17 males, 1 female, and an age range from 30 to 50 years old. Their participation in individual interviews and user tests was detailed in Table 1. They were the actual construction project managers, installation supervisors, and engineers of the studied pilots from KONE. Their experience in the elevator industry varied, with the minimum of three and the maximum of 20 years. However, all were highly competent with the machine room planning process and aware of its technical aspects. It is important to note that all participants had none to little experience with VR. For some, exposure to VR occurred via commercial demonstration at shopping malls and social media. None of the participants had used VR for working purposes. In the paper, participants were referred to as manager or engineer in general to avoid the risk of revealing their identity.

**Table 1.** The role of each participant and how they were involved in the study. Short interview took 30 min to conduct while long interview took one hour.

| Pilot Project [1] | Role of Participant | Individual Interview | | User Test |
|---|---|---|---|---|
| | | Short Interview | Long Interview | |
| 1 | Project Manager | - | x | x |
| | Customer Solution Engineer | - | x [2] | - |
| | Installation Supervisor | - | x [2] | x |
| | Project Director [3] | - | - | x |
| | Customer Solution Engineer | - | x | x |
| 2 | Maintenance Manager | - | x | x |
| | Quality Control Engineer | - | x | - |
| | Customer Solution Engineer | - | - | x |

**Table 1.** *Cont.*

| Pilot Project [1] | Role of Participant | Individual Interview | | User Test |
| --- | --- | --- | --- | --- |
| | | Short Interview | Long Interview | |
| 3 | Customer Solution Engineer | - | x | x |
| | Project Engineer | - | x | - |
| | Construction Manager | x | - | - |
| | Installation Manager | x | - | - |
| | Installation Supervisor | - | - | x |
| | Project Manager | - | - | x |
| 4 | Customer Solution Engineer | - | x | - |
| | Project Manager | x | - | - |
| | Installation Supervisor | x | - | - |
| | Field Support Engineer | x | - | - |

[1] To anonymize the identity of participants, the pilot projects were listed as number. [2] Interviews were conducted in written format via email due to a language barrier. [3] As the role was factually similar to project manager, we refer to the participant later as project manager for simplicity and anonymity.

### 3.1. Individual Interviews

There were 14 individual interviews in total. The semi-structured interview method was used in both the first and last stage of the research procedure. This method provides flexibility to adjust and formulate interview questions based on the interviewee's response to deepen the conversation [65]. Due to different availability, two interview types were conducted: a 30-min-short-interview (5 interviews) and a 1-h-long-interview (9 interviews). Two interviews were conducted in a written format due to the language barrier, following the long interview structure. Questions and responses were exchanged via email.

In both interview types, three categories of questions were asked: (1) current challenges in the machine room planning process, (2) their assumption of VR in general, and (3) their perception on the use of VR in the process such as expectation, limitation, adoption requirements and specific use cases. The longer interview type provided more time to deep dive into the consequences of the encountered challenges such as detailed story on how they were resolved and their impact on the project (e.g., extra cost, time, customer satisfaction, etc.), as well as their viewpoint on VR as a collaboration tool.

### 3.2. Multi-User VR Environment Design

The multi-user VR environment was created to suit the need of remote collaboration during the machine room planning process. The purposes of the system were to provide access to 3D models of the construction projects and enable several users to interact with each other and the models simultaneously. The environment was based on DesignSpace—a commercial cloud-based VR design software utilizing Unity game engine. The development was in collaboration with its developer, 3DTalo. Hence, the testing VR environment had all the features from DesignSpace with an addition of the new cabling tool (Figure 2). The presence of users was represented through avatars. Users could move instantly to a desired landing spot via teleporting. Another navigation method was flying, which allowed users to move freely without being attached to the floor level. Tools extensively used in the user test were the measurement tool to provide 1:1 scale measuring, the drawing tool to draw 3D cube or freehand drawing, the cabling tool to present electrical conduits, and the camera tool to capture the virtual scene. Moreover, the VR environment allowed users to interact with the 3D models without VR headset via desktop mode. However, the level of immersion and interaction was limited in such use. For example, cabling tool was not implemented in the desktop mode.

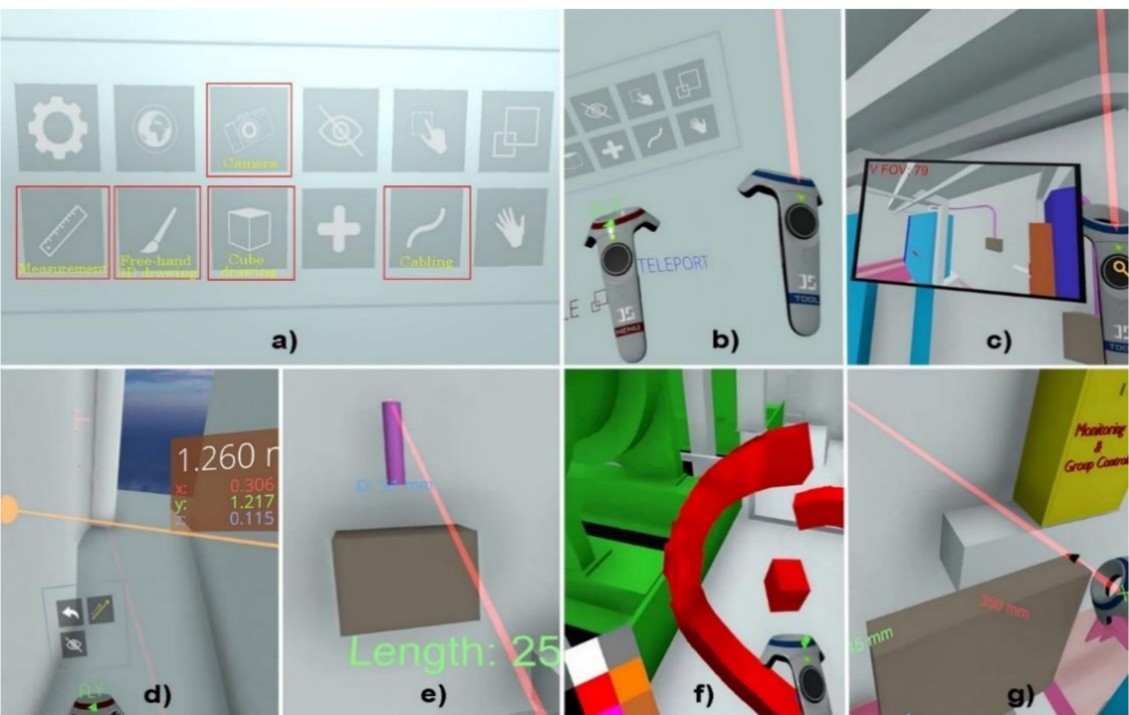

**Figure 2.** The VR environment: (**a**) Menu with most frequent tools highlighted in red; (**b**) Controllers with Navigation options (Teleport and Fly) on the left one; (**c**) Camera tool; (**d**) Measurement tool; (**e**) Cabling tool; (**f**) Free-hand 3D drawing tool; (**g**) Cube drawing tool.

To create the environment, BIM content of the machine rooms from selected pilots were delivered in RVT format (Revit). 3D models under the FBX format were extracted using the default exporting in Revit. Next, manual editing was conducted using Blender to regain texture lost during the exporting. The FBX files were then converted to the VR environment using DesignSpace. The new cabling tool was designed to replicate the exact specification in KONE and developed by 3DTalo in collaboration with one engineer participant.

### 3.3. User Testing

The aim of our user tests is to identify the potential benefits of VR in the industry, not the usability of the VR environment. Each user test consisted of the following sections: onboarding, performing test and semi-structured group interview. There were nine participants and three user tests in total. Initially, onboarding took place either one day prior or at the same day of the test. Participants had 30 to 60 min, depending on their availability, to learn about VR. Three VR environments were created based on the three pilot projects, one for each user test. Every testing session had participants from two countries: one from the pilot of the testing VR environment and one from a different country.

The setup of each user test was illustrated in Figure 3. The test was facilitated remotely by the facilitator. Participants, the VR users, from each country gathered and performed testing at their designated location. There was also one assistant at each testing site to provide technical support. Observers joined the session from their respective remote location. The communication between testing locations, the facilitator, and observers was conducted via Microsoft Teams. The camera of two testing locations were always on to allow remote observation. In addition, the view of respective participants inside the VR environment of the testing pilot was screenshared (Figure 4). Details on the hardware used in the study were summarized in Table 2.

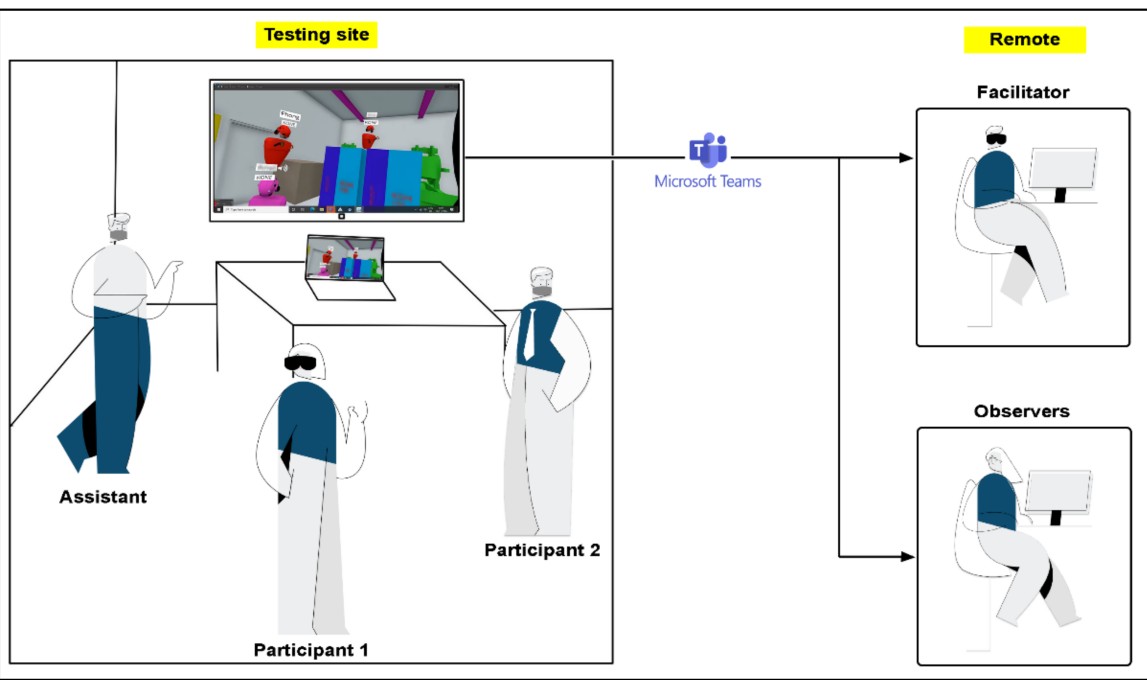

**Figure 3.** User test set-up. Each session had two testing sites and more than one observer. Illustrations were taken from http://www.getillustrations.com/ (accessed on 10 September 2021).

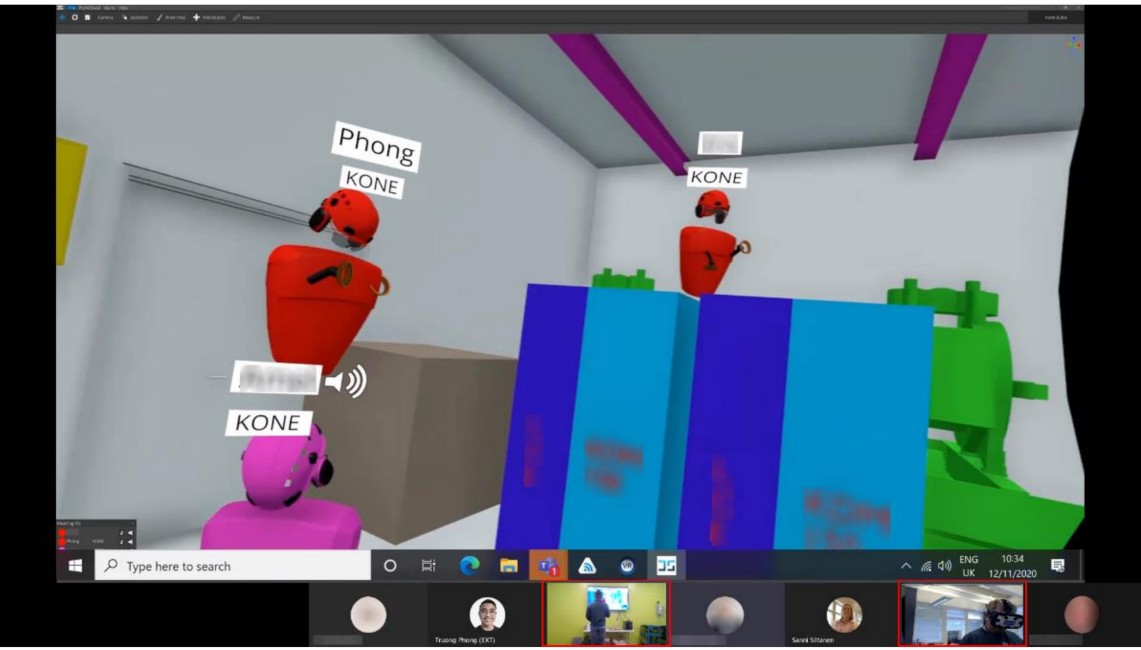

**Figure 4.** Microsoft Teams setup for remote observation.

**Table 2.** Hardware used in user tests.

| Testing Session | VR Headset | Computer System |
|---|---|---|
| 1 | Oculus Rift S<br>HTC Vive | CPU Intel® Core™ i7-4720HQ @ 2.6 GHz, 8 GB RAM, NVIDIA® GeForce® GTX 980 M<br>CPU Intel® Core™ i7-8700K @ 3.70 GHz, 32 GB RAM, NVIDIA® GeForce® RTX 2080 Ti |
| 2 | HTC Vive Pro + Wireless adapter<br>Oculus Rift S | CPU Intel® Core™ i7-8700K @ 3.70 GHz, 32 GB RAM, NVIDIA® GeForce® RTX 2080 Ti<br>CPU Intel® Core™ i7-10750H @ 2.6 GHz, 16 GB RAM, NVIDIA® GeForce® RTX 2070 |
| 3 | HTC Vive Pro + Wireless adapter<br>HP Reverb | CPU Intel® Core™ i7-8700K @ 3.70 GHz, 32 GB RAM, NVIDIA® GeForce® RTX 2080 Ti<br>CPU Intel® Core™ i7-8750H @ 2.2 GHz, 16 GB RAM, NVIDIA® Quadro P1000 |

At each testing location, participants took turns to perform given tasks in the VR environment, which lasted 20–30 min. While using VR, users were encouraged to utilize the think aloud method, meaning that they verbally described their actions and thoughts. The tasks resembled the actual ones in the machine room planning process that participants often worked on with other tools. All the tasks belong to two categories: (1) design review and (2) layout planning of the elevator machine room. In design review, participants were asked to evaluate the accessibility, maintainability, and installability of the machine room layout, respectively. Initially, they examined if access to the machine room from the outside and critical components for operation inside were restricted. Next, users inspected whether there was enough clearance space for future maintenance tasks on some components such as the motor, drive, and controller cabinets. Then, participants were asked to identify potential challenges for the installation of the machine room. All interferences were marked with the Free-hand 3D drawing tool and captured with the Camera tool. The measurement tool was used extensively along with different modes of navigation and object movement in the VR environment for this task category. In layout planning, layout modification by placing missing components (e.g., main switch) using Cube drawing tool and routing certain electrical conduits with Cabling tool was conducted. As the goal was to identify the real-life benefits of VR, participants had the opportunity to propose and perform tasks deemed beneficial for their work.

Finally, a semi-structured group interview was conducted after each user test. It lasted about 45 to 60 min. Participants shared their feedback on their experience. Besides, they were also asked to provide their opinions regarding the use of VR in machine room planning such as benefits, use cases, adoption requirement, etc.

## 4. Results

In this section, the results from the individual interview before testing and the group interview after testing as well as our observations in the user test are presented.

### 4.1. Challenges and Associated Cost in Machine Room Planning Process

Complex communication flow and means with internal and external stakeholders is the critical challenge in the process. 2D drawing, deemed as "lacking feeling of size and space" and "too simple that does not contain enough information" by all participants, is predominantly used to communicate design. The use of BIM is limited and depends on client's request. Missing information caused incorrect material estimating, which results in up to 30% of excess materials. One of the managers reported having underestimation of equipment and room size, which caused inappropriate planning for site logistics and required 8–9 extra weeks to resolve the issue. Limited spatial understanding issue persists even with the help of BIM tools.

Especially in elevator machine room planning, space constraint is typical and experienced in 30% of high-rise projects. The room size is often limited as it does not generate commercial value. However, various equipment from many stakeholders needs to be placed inside, making the design process more challenging. In addition, determining trunking and electrical route for the varied types of cables in such tight environments is "the biggest concern", as referred by all engineers, managers, and supervisors. They noted that consequences of machine room design error could not be recognized immediately and

were often experienced in the installation phase or much later during operation. The result often associates with considerable expenses to resolve, causing project delay and lower customer satisfaction. Previous cases tend to cost over several tens of thousands of euros to re-route electrical conduits due to the unexpected additional equipment that are not included in the official layout drawings.

### 4.2. User Experience and Challenges Encountered with the VR Environment

Participants were captivated by the experience of performing their daily task in VR. Some even stayed longer to explore the tool. The managers expressed their willingness to invest in the solution. Even though all participants had none to very limited experience with VR, their assumptions about its benefits before testing match with findings yielded after. All the features in the VR environment were deemed useful.

Some challenges were encountered when conducting the studies. Firstly, during the first user test, two participants experienced motion sickness. Both took around 5 min of resting to recover and expressed their unfamiliarity with VR might be the cause. Secondly, the VR environment generation process in this study required manual work and involved many file conversions. Participants who helped researchers in the process considered it impractical in the real-world scenario. They demanded a straightforward approach from BIM to VR without extra effort. Thirdly, during one user test, synchronization issues between participants in different countries occurred, which resulted in disruption to collaboration. Unstable internet connection was suspected to be cause. Moreover, participants remarked on some general usability issues of the VR environment. They often forgot the placement of the tools but remembered how to use them once reminded thanks to the VR introduction session. Participants also suggested improving the intuitiveness by decreasing the number of steps taken to perform an action in a tool. Finally, participants indicated the inability to view the overall layout in VR as via 2D drawings. They proposed to include immediate access to 2D layout drawings inside the VR environment.

### 4.3. The Benefits of VR from the User Perspective

All participants indicated the 1:1 scale visualization in VR to enable an immersive perception. Equipment size can be realistically perceived, which was considered better than only imagining in 2D drawings or via BIM models. Participants found a high level of spatial comprehension and considered it as the most critical property of VR. Some realized that the machine room "looks smaller than they imagine" and repositioning of equipment might be needed. The ability to measure and perceive machine room layout accurately in 1:1 scale helps improve the participant's confidence in design review. One manager and one supervisor indicated a great satisfaction in their work. Their design ensures enough space for electrical conduits behind the motors, which usually cannot be tested with other tools. Using VR as a measurement proof (Figure 5), they emphasized the extra confidence helps them gain more control over site logistic planning to optimize labor and cost.

Most importantly, a high level of immersion was observed during the user tests. Users located in different countries indicated the feeling of being together in the same construction site. One engineer noted "having everybody look at the same thing" simplified the need for further explanation, while one manager indicated the higher attention gained from participants compared to that in conventional online meeting. One participant even tried to poke another user for blocking his way. Some attempted to sit on the floor to view beneath the machine. Besides, the ability to communicate using non-verbal gestures and daily verbal expressions in a face-to-face setting reduces frictions in remote communication. Verbal clauses such as "I lost you", "where he is standing", and deictic expressions like "here" and "there" were deemed to make remote communication easier. One engineer utilized the testing time to demonstrate his plan with his manager (also a participant) and found it "much easier than showing this in Revit" as well as noted that it took less time than he expected. Overall, all participants concluded that the collaboration through multi-user VR is superior to the current way of using teleconferencing software.

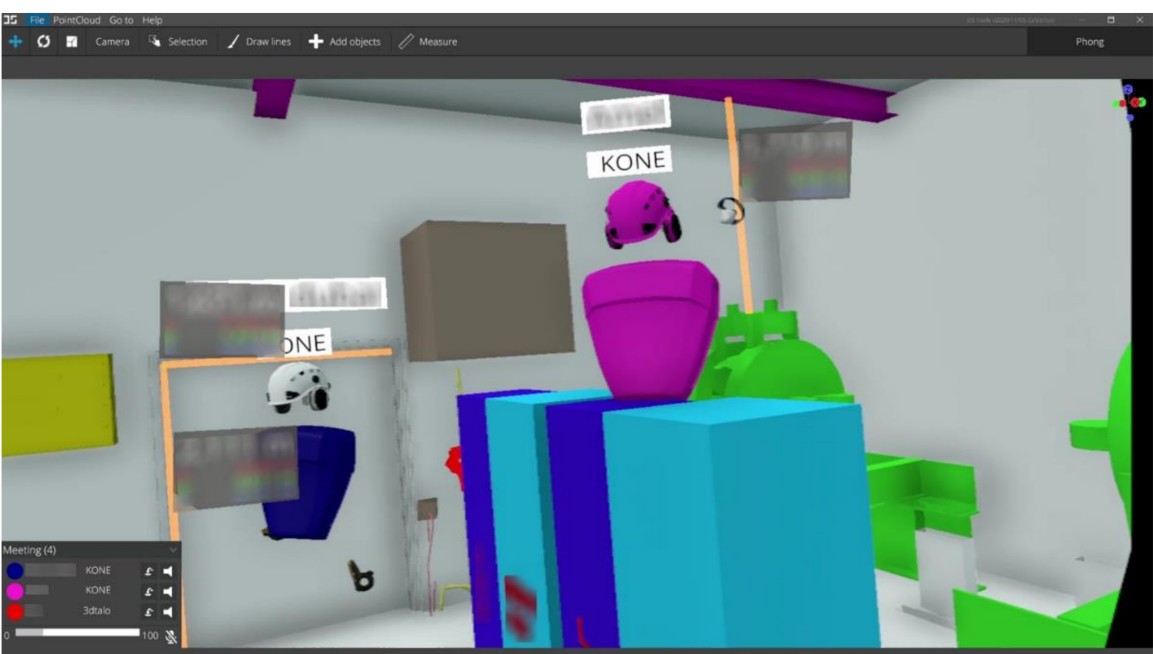

**Figure 5.** A scene inside the VR environment in one user test. Two users used the measurement tools to discuss if there was enough clearance space for installation and future maintenance activities of a motor.

VR is also considered more intuitive for machine room planning by all participants. Users have the freedom of movement to review design from different angles without being restricted to a certain viewpoint as in conventional 2D drawing or BIM tool. They also remarked on greater interactivity than presenting 3D BIM models on a 2D display. One engineer emphasized this benefit comes in useful especially when reviewing irregularly shaped objects. Drawing cable in VR was also found easier as one engineer noted such tasks will require complex tool manipulation in Revit.

VR helps users to easily identify potential constraints and challenges in design. Throughout testing, many critical issues in the layout design were realized. Participants indicated they might cause safety hazards for the installation and maintenance activities on-site. Some even utilized the user test to adjust the layout by reorganizing and adding components to the plan.

The concern of whether the benefit of seeing BIM in VR is significant compared to viewing it on a 2D display, raised by one participant before the user test, was resolved by the time testing completed. The participant recognized the noticeable improvement in BIM coordination process efficiency after using VR to perform design review of his own project with his manager during the test.

### 4.4. The Benefits of VR from the Business Perspective

Participants remarked that applying multi-user VR helped companies to save time and cost. They considered VR to be most economically beneficial in the design and installation phase through some use cases such as BIM coordination meeting, layout design review, and site logistic planning. Coordination meetings within the companies and with external stakeholders can be shortened to 30 min one, replacing the need for two or three longer sessions. Real scale measurement and the ability to plan the construction ahead provide more accuracy in logistic planning and material estimation. Hence, excess materials and extra costs to resolve unexpected challenges can be saved.

Most importantly, one manager emphasized the substantial timesaving to perform design review with clients. He pointed out acquiring design approval from clients was critical and often lasted several months. This process in his current project has taken place for four months and still required longer time to complete at the time of conducting this research. If prolonged, it could be a roadblock as proceeding tasks such as material

purchasing could not be initiated. Using multi-user VR such as in this study, he could immediately demonstrate every adjustment which can also be immediately assessed by the client. "We can get our machine room signed off in hours, not months", said the manager.

The application of VR can reinforce KONE brand as the leader in technological advancement. One manager remarked on the story of how he used 4D BIM to demonstrate a complex installation process to the main contractor. It has left a powerful impression on the customer who later asked other companies to provide such demonstration. Similar effects can be projected with VR, hence enhancing the company's market competitiveness and attracting more clients.

## 5. Discussion

In this study, we used semi-structured interviews and VR user tests to identify the actual benefits of VR in the machine room planning process. The results showed that VR improved planning accuracy, collaboration, and satisfaction for the users as well as brought significant economic benefits to AEC businesses. The findings in this study are highly relevant in the industry context by involving real-life professionals to perform testing with their respective projects. This study stands out from the existing work by conducting the virtual collaboration in a truly remote setup, whereas previous studies often tested such collaboration in a co-located setting.

### 5.1. Both Users and Business Can Benefit Significantly from VR in the AEC Industry

The main challenges in the current elevator machine room planning—inefficient communication and predominant use of 2D drawings—have been encountered in other construction projects [1,2,19,24,50,52]. All benefits found in this study correspond to findings of many others that investigate single-user VR [18,52,54,66] and multi-user VR [3,19,20,67,68] in the AEC industry. Users recognize the strong sense of presence, the flexibility in viewing immersive and full-scale environments, as well as the ease of communication as the key benefits of VR. They also serve as the foundation that generates other benefits. Ultimately, the use of collaborative VR can benefit both users and business in construction projects.

From the user perspective, multi-user VR is superior to conventional BIM tools, teleconferencing software, and the combination of both in the current remote work setting. Collaboration is deemed significantly more efficient due to the strong sense of co-presence with the use of non-verbal cues and daily verbal communication (e.g., deictic reference) [3,5,15,55]. Moreover, improved confidence and accuracy in design review is a critical merit. Participants consider it to not only bring high work satisfaction but also gain more control in the installation phase. Our results support the conclusions presented in previous work [1,2] that multi-user VR could solve critical communication challenges in the AEC industry. As remote collaboration is characteristic of construction projects [6,7], multi-user VR will remain beneficial even in the post-COVID world. Nevertheless, major challenges encountered during designing and testing VR environments in this study can significantly affect the user's willingness to adopt VR. Most importantly, the manual work required to address the interoperability between BIM and VR remains a critical technical limitation [1,25]. Adopting VR means introducing a new step to the workflow; hence its implementation should not require extra effort [23,25]. The problem leads to another issue which is the inability to retrieve data from BIM in VR. It can negatively affect the quality of remote communication as the decision-making process in the AEC industry is highly data-driven and not all data is always available [36].

In the business context, reduction in time, waste and extra cost is the main benefit as also reported by other studies [8,17,22,57,59]. Our study clearly demonstrates the quantified economic gain, addressing the knowledge gap of the actual VR benefits in the business context [21,24,25]. Complex construction projects such as the ones in this study often last two to four years with the first year dedicated for the design process. Hence, the reduction of several months within the design phase to acquire client design approval is a significant time saving for any construction projects. Moreover, high fixed costs and

up to 28% of cost overruns [69–71] often cause low profit margin in large construction projects. Hence, despite that the total project cost is often over several million euros for specialty contractors, several tens of thousands of euros saving provides companies with great flexibility to stretch their budget. In addition, revenue gains through other sources such as winning more contracts through enhancing market competitiveness [8,59] are also notable. However, these economic benefits of VR must justify its implementation cost, which was not investigated in this study. This issue remains the critical research gap for the industry decision-makers to favor using VR in the current workflow [21,24,25].

### 5.2. Industrial Implications

The following list of suggestions aims to help companies that consider adopting VR in their workflow at an early stage:

- **Providing opportunity to experience VR**: Increasing awareness is critical to enhance VR technology acceptance [19]. This study shows that conventional reporting such as written reports, videos, presentations, etc. is not effective enough. Participants were well-aware of the benefit of VR but did not consider its adoption. Hence, everybody should be encouraged to experience using VR to raise awareness and consideration to use it.
- **Identifying critical use cases**: The results suggest three following use cases that all construction projects can apply to maximize the benefit of VR: coordination meeting, layout design review, and site logistic planning. However, each project and company may have different needs that require defining other use cases to balance the cost-benefit ratio of using VR.
- **Developing suitable VR software**: Necessary features should be identified and tailored as different use cases and user needs may require specific functionalities [19,23,50]. Most importantly, the software must accommodate design-to-VR process as well as high data synchronization between BIM authoring tool and VR [1,25,36].
- **Determining application of VR on a case-by-case basis**: Not all projects should utilize VR, as its benefit might not justify the required investment. Supporting the recommendation by Liu et al. [18], companies should consider whether other stakeholders in the market and the project team have the competency and resources to accommodate the use of VR. BIM usage is required to produce accurate virtual environments. Besides, the engineering complexity of the project should be considered since only large and complex projects are recommended in the early stage [18,27].
- **Executing a robust implementation:** Powerful VR hardware should be available at branch level for immediate access. Adequate training and its materials should be developed and provided for anyone in need.

### 5.3. Limitation and Future Work

A limitation of this study is the narrow focus on the use of VR in a single instance in the planning process. Despite the actual benefits presented in the industry, future work should study VR application throughout the project's life cycle to explore its benefits as well as implementation challenges in different stages. Another limitation of the study is the different on-boarding duration that might influence how users perceive the intuitiveness of VR. Participants with shorter training time tend to struggle more frequently than those with longer learning time. As no participants have used VR before, it is worth noting that they only had one session to learn using VR for the user test. Evidence [72–74] has shown that more frequent use of VR enhances user ability to easily utilize VR and reduces motion sickness. Hence, similar future studies can increase the number of VR exposures before testing [74]. This should not affect the results as we studied the benefits of VR rather than the general usability.

To support the wide adoption of VR in the industry, it is critical to address the concerns of implementing VR from a business and user perspective. Future study should investigate the actual implementation cost and how it compares to the benefits of VR. This is extremely

important for decision made in favor of adopting VR by the upper management [21,24,25]. Moreover, our findings suggest the interoperability between BIM and VR is the critical technical limitation from the user viewpoint. Hence, future development of VR should prioritize the robust two-way communication between BIM and VR to streamline the implementation process within construction projects.

## 6. Conclusions

With the shift towards remote working, collaborative VR has been discussed to address the current communication issue caused by the predominant use of 2D drawings in the AEC industry. Research has identified its critical merits; however, there is still a knowledge gap between academia and industry about the actual benefits of the technology in the business context. Such gap results in the slow VR adoption rate in the AEC industry. To fill this gap, this study aims to understand the real-life benefits of multi-user VR from the user and business perspective via one AEC scenario—high-rise elevator machine room planning. Four real-life high-rise projects in different countries were involved.

The results indicate that VR is more intuitive for planning compared to conventional tools, which increases work efficiency, accuracy, and satisfaction from the user perspective. Multi-user VR enables a robust remote collaboration to address the communication issue in the industry. Significant real-life economic benefits thanks to the reduction of time in acquiring client design approval and of cost to resolve design errors are identified. Our findings are highly industry-relevant by involving real-life professionals in realistic scenarios. This study also stands out due to the true remote collaboration setting while existing studies often have co-located setup. To further help companies to successfully adopt VR in the early stage, we provided a list of suggestions on the implementation practicalities. Two future research directions from the business and user perspectives were proposed: (1) defining the true cost-benefit ratio of using VR and (2) developing a robust communication between BIM and VR, respectively.

**Author Contributions:** Conceptualization, P.T., S.S. and K.H.-O.; methodology, P.T., S.S., K.H.-O. and R.T.; software, P.T. and P.B.; formal analysis, P.T.; writing—original draft preparation, P.T., S.S., P.B. and R.T.; writing—review and editing, P.T., K.H.-O. and S.S.; visualization, P.T.; supervision, S.S. and K.H.-O.; All authors have read and agreed to the published version of the manuscript.

**Funding:** This research was funded by Business Finland under the KEKO Smart Building Ecosystem Project (https://kekoecosystem.com/ (accessed on 10 October 2021)).

**Acknowledgments:** We would like to thank Peter Eagling, Daniel Maughan and Amelia Veronica for their input and help in organizing the user tests. We also thank Iiro Naamanka for his help with DesignSpace during the VR environment development.

**Conflicts of Interest:** The authors declare no conflict of interest.

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
