# Peer review of "Multi-User Virtual Reality for Remote Collaboration in Construction Projects: A Case Study with High-Rise Elevator Machine Room Planning"

_electronics, doi:10.3390/electronics10222806_

Round 1
Reviewer 1 Report
Manuscript ID MDPI electronics-1458499
Multi-user Virtual reality for remote collaboration in construction
The authors present a study investigating the suitability of VR technology for remote collaboration in construction. The topic is important for the field. All the references are referenced in the manuscript.
The authors can consider the following suggestions for improvement:
- More information should be given about the study participants (age, gender, technical background, VR experience).
- User scenarios, tasks and evaluation procedure should be described in more detail. What exactly were participants’ tasks?
- Which participants were involved in user testing and which in interviews, as this is not entirely clear from the manuscript.
- Were there any other methods used to assess the user experience, usability, immersion/presence of the proposed solution?
- Were there any VR sickness effects observed during the session?
Reviewer 2 Report
In this study the authors present a timely and interesting research. The structure of the study is logical.
The literature review is thoroughly conducted: the article has a large number of references relating to this field of research and virtual reality is well-presented.
Section 3.1: One of the most important parts of the study were the interviews. Therefore, this should be elaborated on more. For example, what were the questions on the two interview types? A table should be included as well where the interviewee data should be shown (number per profession, average age, number per gender).
Figure 2: According to the template, the caption of subfigures should be stylized as (a) text about a; (b) text about b; (c) text about c; (d) text about d; (e) text about e; (f) text about f; (g) text about g.
The language used in the study is fine, only a minor spellcheck is required.
The references are in the correct format of the journal.
